# Audiovisual Integration for Saccade and Vergence Eye Movements Increases with Presbycusis and Loss of Selective Attention on the Stroop Test

**DOI:** 10.3390/brainsci12050591

**Published:** 2022-05-03

**Authors:** Martin Chavant, Zoï Kapoula

**Affiliations:** 1IRIS Laboratory, Neurophysiology of Binocular Motor Control and Vision, CNRS UAR 2022, University of Paris, 45 rue des Saints-Pères, 75006 Paris, France; chavantmartin@gmail.com; 2Orasis-Eye Analytics and Rehabilitation, 45, Rue des Saints-Pères, 75006 Paris, France

**Keywords:** audiovisual integration, presbycusis, aging, saccade, vergence

## Abstract

Multisensory integration is a capacity allowing us to merge information from different sensory modalities in order to improve the salience of the signal. Audiovisual integration is one of the most used kinds of multisensory integration, as vision and hearing are two senses used very frequently in humans. However, the literature regarding age-related hearing loss (presbycusis) on audiovisual integration abilities is almost nonexistent, despite the growing prevalence of presbycusis in the population. In that context, the study aims to assess the relationship between presbycusis and audiovisual integration using tests of saccade and vergence eye movements to visual vs. audiovisual targets, with a pure tone as an auditory signal. Tests were run with the REMOBI and AIDEAL technologies coupled with the pupil core eye tracker. Hearing abilities, eye movement characteristics (latency, peak velocity, average velocity, amplitude) for saccade and vergence eye movements, and the Stroop Victoria test were measured in 69 elderly and 30 young participants. The results indicated (i) a dual pattern of aging effect on audiovisual integration for convergence (a decrease in the aged group relative to the young one, but an increase with age within the elderly group) and (ii) an improvement of audiovisual integration for saccades for people with presbycusis associated with lower scores of selective attention in the Stroop test, regardless of age. These results bring new insight on an unknown topic, that of audio visuomotor integration in normal aging and in presbycusis. They highlight the potential interest of using eye movement targets in the 3D space and pure tone sound to objectively evaluate audio visuomotor integration capacities.

## 1. Introduction

Presbycusis is an important health issue, even more so in the context of the global aging of the population. In 2011, a study in the US [1] found that 63% of the population above 70 years have at least slight hearing loss. The well-known consequences of presbycusis are an increase in the hearing threshold, a poorer-frequency resolution, and comprehension in silence and in noise [2]. Nevertheless, other consequences of presbycusis have been studied over the last two decades, notably showing associations with dementia [3,4], depression [5], and cognitive deterioration [3,6,7,8,9], independently of age. Thus, the consequences of presbycusis appear to a much greater extent than just hearing issues and seem to affect more central cortical aspects.

Multisensory integration (MI) is defined as an interactive synergy among the senses [10]: Different sensory modalities, with respect to a certain proximity in time and space [11,12], are integrated into the same neuron population. It provides a better perception of a multimodal stimulus if its sensory modalities are congruent. Effective multisensory integration processing relies on the proper functioning of (i) the cortical and subcortical circuits on which it depends (top-down processes) and (ii) the various peripheral sensory receptors and their neural paths (bottom-up processes). Among the cognitive factors affecting MI are, notably, semantic congruence [13] and selective attention [14,15].

Audiovisual integration (AVI) is probably one of the most-used kinds of multisensory integration. However, there is currently little knowledge concerning the effect of presbycusis on audiovisual integration. Studies on the influence of hearing loss on MI deal mostly with profound, long-standing hearing loss (as opposed to presbycusis, which is a more slight and recent loss of hearing). Two review studies report evidence of multisensory reorganization with old and profound hearing loss, including the activation of neurons in the auditory cortex for non-auditory information [16,17]. There is also substantial literature on the multisensory consequences for persons with cochlear implants, showing that audiovisual integration in these populations has been strongly impacted by multisensory cortical reorganization [18,19,20].

The few existing studies dealing with the behavioral effect of presbycusis on audiovisual integration use various paradigms. Some of them assess audiovisual speech performances [21,22], while others use a multisensorial distractor paradigm [23] or the McGurk illusion, which consists of presenting incongruent audio-visual syllables leading to an illusionary percept (e.g., an auditory “ba” and a visual “ga” lead to the perception of the “da” sound) [24,25]. Although their results are contradictory (see discussion), they would indicate multisensory consequences of presbycusis. This is supported by recent electrophysiological studies showing that multisensory cortical reorganization in the auditory cortex is also possible following mild to moderate hearing loss [26,27].

In view of the limited number of studies, it is difficult to appreciate the potential effect of presbycusis on audiovisual integration. A plausible hypothesis would be the degradation of audiovisual integration with presbycusis. Indeed, as mentioned above, audiovisual integration is related to the proper functioning of hearing but also to certain cognitive processes, and presbycusis is associated with hearing and cognitive decline. However, in relation to that topic, important literature on the effect of age on MI depicts a more complex mechanism than it seems. Recent reviews indicate that multisensory benefits would be greater for older populations than for younger populations [12,28,29,30,31]. Several hypotheses have been formulated to explain the mechanisms behind this improvement of MI with age [28], but the evidence is still lacking in favor of one or the other hypotheses. Thus, on the one hand, presbycusis could degrade audiovisual integration, and on the other hand, one could expect an improvement as a compensatory mechanism.

Given the importance of audiovisual integration and the prevalence of presbycusis, learning more about these two phenomena is of high importance, and research on this topic might also extend our knowledge of aging. The purpose of this study is to develop this issue using oculomotricity. The measurement of eye movements, especially via saccade latencies, is a good tool to evaluate multisensory effects. A well-known beneficial effect of combining matching visual and auditory modalities is the reduction of the reaction time. It has been proven for manual reaction time [32] as well as for the latency of saccade eye movements [33,34,35,36,37,38,39]. However, while saccade latency is the measure that has been used most in this field, it is also interesting to use other oculomotor features and other types of eye movement. Studies have also shown an improvement in saccade accuracy with audiovisual targets [37,39]. Most importantly, Kapoula et al. [40] were the first to study the differential effect of sound on saccades and vergence eye movements in a multiple-case study. They reported a decrease in the saccade latency but an increase in vergence velocity when comparing audiovisual vs. visual targets. Thus, it is possible that the sound acts differently for the two types of eye movements. To what extent this depends on the type of population studied, namely on age, is not yet known.

Thus, the main focus of this study is to investigate the relation between presbycusis, age, and audiovisual integration by assessing the programming and execution of saccades and vergence eye movements toward visual and audiovisual targets. We will study these relationships both in young and elderly populations compared to each other, but also within the elderly population alone, which is the only one presenting presbycusis. The aim is to better understand how the brain handles audiovisual integration when hearing capacities are becoming progressively weaker for an elderly population. Would audiovisual integration be degraded in relatively recent and age-related mild hearing loss? How would it evolve with age? Would it deteriorate because of the sensory loss, or oppositely, would it improve thanks to a compensatory mechanism? 

We also decided to evaluate the relationship between the Stroop test and audiovisual integration. The Stroop test [41] is a gold standard to assess the selective attention and inhibition capacities, which are degraded with age [42,43,44,45]. This test allows us to characterize the cognitive abilities of our elderly population. Furthermore, it is relevant to associate it with audiovisual integration, as selective attention has been identified among the top-down cognitive processes that can modulate MI. More precisely, restricted attention to a particular sensory modality diminishes the multisensory enhancement given by the addition of another congruent sensory modality [14,15,46], and adults with attention deficits are more easily distracted by incongruent sensory stimuli [47,48,49].

## 2. Materials and Methods

The study was conducted in accordance with the Declaration of Helsinki and approved by the Ethics Committee “Ile de France II” (N° ID RCB: 019-A02602-55, approved the 3 October 2020).

### 2.1. Participants

An elderly group (EG) and a young group (YG) were tested. The young group was composed of 30 participants aged between 21 and 30 years (mean 25.3 ± 2.68). They were mostly students working in neighboring laboratories. The elderly group was composed of 69 participants, aged between 51 and 84 years (mean 66.7 + 8.4). They were essentially recruited on the RISC (relai d’information des sciences cognitives, France) platform of the CNRS and were autonomous. We excluded from the study persons with hearing loss wearing hearing aids, conductive deafness, genetic deafness, who were frequently exposed to loud noises or took ototoxic drugs, persons with visual pathologies (e.g., AMD, non-operated cataract, or glaucoma), persons with ocular motor abnormalities (e.g., strabismus, ptosis, etc.), and persons taking drugs likely to affect sensory and motor functions. All this information is verbally requested from the participants. Among the elderly participants, 5% were treated for diabetes, 17% were treated for blood pressure issues, none had renal failure, and 14% had vascular issues (60% of which were treated). Thus, the participants of the elderly group are considered to be a normal, representative aging population. Moreover, the exclusion criteria applied ensured that the potential hearing losses observed in the old group are due to age. Informed consent was obtained from all of the participants after the nature of the procedure had been explained.

### 2.2. Hearing Tests

A professional audiometrist was in charge of performing two kinds of audiometry, in a sound booth calibrated cabin, with an audiometer of the brand Interacoustics (Middelfart, Danemark),model AD639. The outer ear canals of all the participants were all checked before the hearing tests. Hearing tests were performed in 62 of the 69 elderly participants and 19 of the 30 young participants; as the audiometry tests were performed outside of the laboratory, some of the participants were no longer available for this second session.

#### 2.2.1. Tonal Audiometry

This test aims to assess the audibility, i.e., the minimum intensity required for a sound to be heard. It was realized, for each ear separately, with the headset TDH-39P. For each ear, we determined the lowest intensity in dB HL needed by the participant to detect the following pure tones: 250, 500, 750, 1000, 2000, 3000, and 4000 Hz. The final score is called the Pure Tone Average (PTA) and represents, for each ear, the mean of all these thresholds. Only the best PTA between the two ears was retained to match the hearing-loss definition of the World Health Organization (WHO). WHO considers hearing loss to be when the best PTA of the two ears is above 20 dB HL [50].

#### 2.2.2. Vocal Audiometry in Silence

This test aims to assess speech comprehension in silence. Word lists with differing intensities are sent to the participant, enabling us to calculate a comprehension score for a given intensity. It was performed with a loudspeaker situated at 1 m in front of the participant. Thus, the two ears were tested simultaneously. The different intensities with which the lists were sent were 70, 60, 50, 40, 30, 20, or 10 dB SPL. The lists used were the Lafon cochlear lists, composed of 17 monosyllabic words of 3 phonemes (51 phonemes) [51]. The comprehension score for each list was calculated with the percentage of recognized phonemes in the list. The final score is called the SRT50 (Speech Recognition Threshold 50%) and represents the intensity required to understand 50% of the phonemes of a list. The SRT50 was estimated by a cross product between the intensities needed for the lists with the score just above and below 50% comprehension.

#### 2.2.3. Vocal Audiometry in Speech

This test aims to assess comprehension among noise. Word lists with a differing Signal to Noise difference (SND) are sent to the participant, enabling us to calculate a comprehension score for a given SND. It was performed with three loudspeakers situated at 1 m distances from the participant: One at their back, one at their right, and one at their left. As in the test for vocal audiometry in silence, the two ears were tested simultaneously. From the two loudspeakers on the side, the word lists were played, and from the loudspeaker at the back, the noise was played. The SND represents the extent to which the speech signal is higher or lower than the noise signal. It is calculated by deducting the intensities in dB SPL of the speech list and the noise (SND = signal intensity-noise intensity). During the entire test, the intensity of the word lists was unchanged, and the SND varied for each new list by changing the intensity of the noise signal. For each participant, the intensity of the word lists was chosen by taking the lower intensity in the vocal audiometry in silence with the best score; for example, if, in the vocal audiometry in silence test, participant A had a recognition score of 100% for the list at 60 dB SPL, 100% for the list at 50 dB SPL, and 82% for the list at 40 dB SPL, then the intensity of the lists for the entire vocal audiometry in noise would be set at 50 dB SPL). The different SND values with which the lists were sent were 0, −5, −10, −15, and −20 dB SPL. As for the vocal audiometry in silence, the lists used were the Lafon cochlear lists. The noise signal used was the “Onde Vocale Globale” (OVG), an incomprehensible babble noise composed of two couples speaking at the same time [52]. The comprehension score for each list was calculated with the percentage of recognized phonemes in the list. The final score is called the SND50 (Signal to Noise Difference 50%) and represents the SND required to understand 50% of the phonemes of a list. The SND50 was in fact estimated by a cross product between the SNDs needed for the lists with a score just above and below 50% comprehension.

### 2.3. Oculomotor Tests

Divergences, convergences, and left and right saccades were elicited with the REMOBI device (patent US885 1669, WO2011073288), a visio-acoustic device developed by our laboratory (Figure 1).

REMOBI is a plane surface where red LEDs are displayed (a frequency of 626 nm, 180 mCd, and a diameter of 3 mm). Each LED is equipped with a buzzer delivering a 2048 Hz pure tone of 70 dB SPL. Participants were seated, and the REMOBI was placed at their eye level. The instructions given to them were to look at the only LED on, as quickly and accurately as possible, and then to maintain fixation as the LED was still on, without moving the head. Thus, the localization and patterns of LEDs allow for testing of the desired eye movements.

During the saccades test, 20 trials of saccades to the right (RS) and 20 trials of saccades to the left (LS) were elicited, randomly interleaved. For each trial, participants first fixated on a central LED, situated at 70 cm in front of them (the same distance from both eyes). The right and left saccades were elicited by lighting a peripheral LED, also at 70 cm, but at 20° to the right or left from the central LED.

During the vergences test, 20 trials of convergence (C) and 20 trials of divergence (D) were elicited, randomly interleaved. All the LEDs were situated in front of the participant (the same distance for the left and right eyes) and varied only in depth. For each trial, participants first fixated on a central LED, situated 40 cm away. The divergence and convergence were elicited by lighting a peripheral LED, situated at either 20 cm (convergence) or 150 cm (divergence) from the participant.

For saccades and vergences tests, the central LED is switched on during a random time between 1200 and 1800 ms. The peripheral LED is lit for 2000 ms. There is an overlap time of 200 ms where the two LEDs are lit at the same time (overlap paradigm). The trials are separated by a blank period of 300 to 700 ms. The total duration of a sequence is approximately 150 s.

### 2.4. The Targets Modality

The saccades and vergences tests are passed for the visual paradigm (Paradigm V) and the audiovisual paradigm (Paradigm AV). In the visual paradigm, the LEDs turn on without activation of their adjacent buzzer. In the audiovisual paradigm, an auditory signal is sent with the adjacent buzzer of the LED, 50 ms before the activation of the LED and for a duration of 100 ms. According to prior studies, such a time delta between the auditory and visual signals (50 ms) is the most effective in terms of shortening the eye movement latency inducing both a warning effect and perhaps better localization of the visual target [35,40].

### 2.5. Eye Movements Analysis

The eye movements are captured with a head-mounted video-oculography device, Pupil Core (Pupil Labs, Berlin, Germany), enabling binocular recording at 200 Hz per eye, using a pupil-tracking mode. The standard Pupil Labs calibration (Pupil Capture) was applied using a target that was presented at eye level with a viewing distance of 1 m. The participant had to fixate on the center of this target and slowly move their head rightward, downward, leftward, and upward, repeating this sequence 3 times [53]. The confidence level was better than 80%. The data acquired are analyzed with AIDEAL software (pending international patent application: PCT/EP2021/062224 7 May 2021).

For saccades, AIDEAL treated the conjugate signal, e.g., the L + R position/2. The amplitude of the saccade movement is measured by defining its onset and offset when the velocity of the movement is above or below 10% of its peak velocity. Practically, this corresponded to values above or below 40°/s (as the peak velocity of 20° saccades is typically above 400°/s). For vergence eye movements, AIDEAL calculates the difference between the two eyes from the individual calibrated eye position signals (i.e., left eye–right eye). The amplitude of vergence is divided into two components, following the double mode control of the vergence model. This model divides the dynamic of vergence into two chronological steps: (i) An initial step of enhanced speed without visual feedback (open-loop) and ii) a sustaining step, slower and driven by visual feedback (closed-loop) [54,55,56]. The initial open-loop component is defined by AIDEAL when the velocity of vergence is above 5°/s. The following closed-loop component is measured by including part of the movement for the next 160 ms. Then different filters on trials are applied. AIDEAL first removed the trials with blinks, then outliers, i.e., values greater than twice the standard deviation. For divergence, 29% ± 12% of trials were excluded for the young participants and 26% ± 20% for the elderly group. For the convergence, 45% ± 14% were excluded for the young participants and 38% ± 17% for the elderly group. For the left saccades, 16% ± 11% were excluded for the young participants and 28% ± 17% for the elderly group. For the right saccades, 9% ± 10% were excluded for the young participants and 24% ± 18% for the elderly group. 

### 2.6. Eye Movements Characteristics Measured

Latency (Lat): Expressed in ms. It represents the time between the activation of the peripheral LED and the initiation of the movement.

Peak Velocity (PVel): Expressed in °/s. It is measured for the total saccade and the initial open-loop component of vergence.

Average Velocity (AVel): Expressed in °/s. It is measured for the total saccade and the total vergence (open-loop component + closed-loop component).

Amplitude (Amp): Expressed in %. It represents the percentage of the amplitude required to reach the peripheral target (20° for saccades, 8.76° for convergence, and 6.5° for divergence).

For each of these characteristics, the variable AVI (AudioVisual Integration) is assigned, representing the variable for the Paradigm AV minus the variable for the Paradigm V: AVI(Lat) = (Lat for AV) − (Lat for V); AVI(PVel) = (PVel for AV) − (PVel for V); AVI(AVel) = (AVel for AV) − (AVel for V); AVI(Amp) = (Amp for AV) − (Amp for V).

### 2.7. Stroop Test

The original Stroop test was created in 1935 by J.R. STROOP [41], and many variations have been created since. A Stroop test aims to assess inhibition and selective attention capacities.

The French version of Stroop Victoria was used in the current study [57]. It is composed of three parts. In each part, participants enumerate as quickly and accurately as possible the color of 24 items (6 lines of 4 items) presented on a sheet of A4 paper. In the first part, called the Dot condition, items are dots. In the second part, called the Word condition, items are irrelevant words (words with neutral meaning). The items of the third part, called the Interference condition, are color words irrelevant to their ink impression (the word “blue” written in red).

The selective attention and inhibition capacities are evaluated by comparing the performances during the Dot and Interference conditions. The Dot condition assesses the baseline capacity of color recognition and enumeration. The Interference condition also assesses these capacities but with the interference of the words. The information given by the instinctive reading of the words is more intrusive than its color and has to be inhibited by the brain.

The score extracted from the Stroop test is called Stroop_I/D and represents the ratio of the time taken to finish the Dot condition to the time taken to finish the Interference Condition. Higher Stroop_I/D reveals lower capacities of selective attention and inhibition.

### 2.8. Data Analyses

The relationships between AVI and age are measured with simple linear regressions and correlations: AVI(Lat)~Age, AVI(PVel)~Age, AVI(AVel)~Age, and AVI(Amp)~Age. These results are presented in the results Section 3.2—AVI and Age.

The relationships between AVI and hearing are measured with multiple regression analysis in order to control age and avoid a potential confounding effect: AVI(Lat)~Hearing + Age, AVI(PVel)~Hearing + Age, AVI(AVel)~Hearing + Age, AVI(Amp)~Hearing + Age. Only the results of the relationships between AVI and hearing are presented in the results Section 3.3—AVI and Hearing. Thus, these results will be independent of age.

For the same reason, the relationships between AVI and Stroop are also measured with multiple regression analysis: AVI(Lat)~Stroop + Age, AVI(PVel)~Stroop + Age, AVI(AVel)~Stroop + Age, AVI(Amp)~Stroop + Age. Only the results of the relationships between AVI and Stroop scores, independent of age, are presented in the results Section 3.4—AVI and Stroop.

## 3. Results

### 3.1. Characterization of the Elderly Group

Before analyzing the AVI relationships, this part characterizes the elderly participants by comparing their results with those of the younger group and with standard results.

Table 1 groups the means and standard deviations of the oculomotor characteristics for each group. Each line refers to an eye movement (D for divergence, C for convergence, LS for saccades to the left, and RS for saccades to the right); values are shown for visual targets and audiovisual targets. Table 2 shows the group means and standard deviations of the hearing scores for each group of participants.

Results show longer latencies for the elderly and for all types of eye movements. The audiometry scores, PTA, SRT50, and SND50 are better for the elderly than for the young, meaning a lower audibility and speech comprehension for this group. The selective attention evaluated by the Stroop test also reflects lower performance for the elderly. Stroop_I/D is 2.05 ± 0.47 for the elderly vs. 1.61 ± 0.30 for the young. All these results are in agreement with prior studies showing a deterioration of hearing with age [1], an increase in oculomotor latencies [58,59,60,61], and a deterioration in the Stroop test with age [43,45]. Our results show all these three deteriorations occurring in the same elderly participants.

Figure 2 regroups the classification of hearing loss and of Stroop results for elderly participants. The hearing loss classification is according to the WHO scale of hearing loss [50] and the Stroop result classification is according to the model built in the study of Bayard et al. [57].

Figure 2A shows that, according to the WHO scale, 46% of the elderly participants had normal hearing, 45% presented mild hearing loss (PTA of the better ear between 20 and 35 dB HL), 6% presented moderate hearing loss (PTA of the better ear between 35 and 50 dB HL), and 1% presented moderately severe hearing loss (PTA of the better ear between 50 and 65 dB HL). Such prevalence is in the normal range and in agreement with prior studies [1]. Figure 2B shows that, given the classification provided by the French Stroop Victoria test, none of the elderly participants were classifiable as presenting cognitive deficiency (none of the Stroop_I/D scores were in the “deficit” category). To summarize, all the results pointed to a healthy aging population.

### 3.2. AVI and Age

The correlations and regression lines between the AVI and age are presented in Table 3.

The four rows assess the relationships between the different AVI and age for (top to bottom) divergences, convergences, left saccades, and right saccades. For each row, the first line shows the results for the whole population (young and elderly participants). The second line shows the results for the elderly group alone. The columns indicate the eye movement characteristics measured (Latency, Peak Velocity, Average Velocity, or Amplitude). Thus, for example, the first row and first column result from the linear regression Divergence AVI(Lat)~Age. The values to focus on are the “a”, representing the slope of the regression line. Their significance level is indicated with asterisks: “***” for a *p* lower than 0.001, “**” for a *p* between 0.001 and 0.01, “*” for a *p* between 0.05 and 0.01, and “.” for a *p* between 0.1 and 0.05. The value “cor” represents the Pearson correlation coefficient.

The results in Table 3 show significant relationships between the AVI for convergences and age. These relationships are also represented in Figure 3. However, they differ in nature depending on the populations studied.

Considering the whole population (young and elderly participants), there is a negative effect of age on audiovisual integration for convergences. This negative effect is reflected in the significant increase in AVI(Lat) for convergences with age (see Figure 3A). In other words, the reduction in convergences latency when adding sound to a visual target is less important for elderly participants than for young participants.

When considering only the elderly participants, there is a positive effect of age on the audiovisual integration for convergences. This positive effect is reflected in a significant decrease in AVI(Lat) with age, and significant increases in AVI(AVel) and AVI(Amp) for convergences with age. In other words, the improvements, related to the addition of sound, in latency, average speed, and amplitude are greater for the older participants of the elderly group.

### 3.3. AVI and Hearing

The regression lines (execrated from multiple regression analyses with age as explanatory results, see Section 2.8—Data Analyses) between AVI(Lat) and hearing are presented in Table 4.

As for Table 3, in the first row are the results for divergence, the second row for convergence, the third row for left saccades, and the fourth row for right saccades. Each of these rows is divided into two lines: The first one presents the results for the whole population (young and elderly participants) and the second one presents the results for elderly participants alone. The columns indicate the hearing score (PTA, SNR50, or SND50). The values to focus on are “a”, representing the slope of the regression line. Their significance level is indicated with asterisks: “***” for a *p* lower than 0.001, “**” for a *p* between 0.001 and 0.01, “*” for a *p* between 0.05 and 0.01, and “.” for a *p* between 0.1 and 0.05.

There are significant effects of PTA and SRT50 scores on saccades AVI(Lat), independently of age, considering the elderly participants alone or for the whole population: AVI(Lat) for saccades decreases as PTA or SRT50 increase, as shown in Figure 4 (however this figure shows the simple regression lines, where the age factor is not controlled). In other words, the reduction of saccades latency, when adding sound to a visual target, increase with the age-related decrease in audibility and speech comprehension in silence.

The regression lines (execrated from multiple regression analyses with age as explanatory results, see Section 2.8—Data Analyses) between AVI(Avel), AVI(PVel), AV(Amp), and hearing are presented in the Appendix A. There are no significant relationships for any kind of eye movement considering the whole population or just elderly participants.

### 3.4. AVI and Stroop

The regression lines (execrated from multiple regression analyses with age as explanatory results, see Section 2.8—Data Analyses) between AVI and hearing are presented in Table 5. The way to read this table is the same as for Table 4, explained in results Section 3.3—AVI and Hearing. The difference is seen in the columns, which represent the different eye movement characteristics (Latency, Peak Velocity, Average Velocity, and Amplitude).

There are no significant effects of Stroop_I/D on AVI(PVel), AVI(AVel), or AVI(Amp), for any kind of eye movement, considering the whole population or just elderly participants.

There is a significant effect of Stroop_I/D on AVI(Lat) for right saccades, and an almost significant effect of Stroop_I/D on AVI(Lat) for left saccades, considering the whole population (young and elderly participants): AVI(Lat) for saccades decreases when Stroop_I/D increases, as shown in Figure 5 (however this figure shows simple regression lines, where the age factor is not controlled). In other words, for the whole population, the reduction in saccades latency when adding sound to the visual target increases with the reduction in inhibition capacities measured with the Stroop.

## 4. Discussion

The major findings of the study are (i) the complex age effects on audiovisual integration for convergence eye movements only, (ii) the increase in audiovisual integration for saccades with age-related hearing loss, and (iii) the improvement in audiovisual integration for saccades for persons with lower selective attention scores as measured by the Stroop test.

### 4.1. Conditions for Improved Audiovisual Integration between Young and Elderly

It is interesting to note that our study shows no improvement in audiovisual integration between the young group and the elderly group. Indeed, most previous studies dealing with the effect of age on audiovisual integration for reaction times show an improvement in the audiovisual benefits in seniors compared to young adults, whether they assessed a color-discrimination task [30] or a simple reaction task [12,31,62,63]. However, studies using spatial discrimination tasks (e.g., follow the orientation of an arrow, look at a location) have more mixed results. The study of Diederich and Colonius [12], as well as that of Zou et al. [64], effectively found greater audiovisual facilitation in the elderly compared to a young population, while the two other studies did not find a difference [65,66]. Perhaps the difference between these studies is related to the activation of visuomotor covert mechanisms in cases where spatial orientation is involved; see, for instance, the theory of the motor basis of visual attention [67]. As the design of the current study involves locating and moving the eyes to the stimuli, the absence of audiovisual integration improvement for elderly participants is in line with some of the previously cited studies. Perhaps, the more complex the task is, the less apparent the component of audiovisual integration (i.e., a ceiling effect).

Among the studies that report an improvement in audiovisual integration, the study of Diederich is the closest to the current study [12], as it also assesses the reaction time of left and right saccades to visual vs. audiovisual targets. Some differences in the study designs of Diederich and Colonius and the current one could explain the differences in results. Firstly, Diederich and Colonius used speakers emitting bursts of white noise for their auditory signal, while we used a pure-tone sound of 2000 Hz. This choice of the auditory signal will impact sound localization. A pure-tone sound of 2000 Hz is harder to localize than a burst of white noise. Indeed, a large spectrum width (as for white noise) increases the accuracy of space localization, and frequencies between 1000 and 3000 Hz have the poorest localization accuracy [68]. Moreover, the elderly population has decreased abilities in sound localization in space [69,70,71]. Thus, a potential enhancement of audiovisual benefits for the elderly could have been compromised by the inability to localize the pure-tone sound during the vergences tests. Good sound localization is important for audiovisual integration, which becomes more important with the spatial proximity of auditory and visual modalities [39]. Another difference between our study and that of Diederich et al. concerns the diode activation paradigm. Diederich et al. used a “step” paradigm, where the target LED light is on at the same time as the fixation LED is switched off, while our study used an “overlap” paradigm, where the target LED light is on before the fixation LED is switched off. The sequence of such events impacts the attentional mechanisms of eye movement preparation [72,73], which are known to be affected by aging [61,74,75]. Finally, it is important to note that the ranges of ages in our study and the study of Diederich are different. Although the means are similar, the range in our study was very large, ranging from 50 to 84 years. In the study of Diederich, the range was more restricted, from 65 to 75 years.

Further research about these differences in study designs could likely help to better understand why audiovisual integration is found to be higher in the elderly than in young persons in some studies but not in others, although such audiovisual integration is always present.

### 4.2. Decrease Followed by Increase in Audiovisual Integration with Age

Considering the whole population (young and elderly groups), the increase in AVI(Lat) indicates the deterioration of audiovisual integration for convergence with age between young and elderly groups. However, considering the elderly participants alone, the decrease in AVI(Lat) and the increase in AVI(AVel) and AVI(Amp) indicate an improvement in audiovisual integration with age within the elderly group.

In other words, the evolution of the audiovisual integration for convergence with age seems to follow a dual pattern: (i) Deterioration for the elderly relative to young adults and (ii) an improvement with age within the senior group. However, no effect of age was found for audiovisual integration for divergences and saccades, either between the young and elderly or within the elderly group.

The first part of this dual pattern (deterioration between young and elderly) is surprising as, to our knowledge, no study to date has found multisensorial deterioration between the young and elderly. However, this is the first study assessing the convergences of eye movement. Convergence is the most complex and fragile eye movement, the first to be affected by age, fatigue, and neurologic problems, which could explain this result. The studies dealing with the aging effect on multisensory integration selected their task to be achievable by young or elderly with a similar effort, which is not the case for convergences eye movements.

However, the second part of this dual pattern (improvement for an elderly population) suggests that multisensory integration can increase with age as a compensatory strategy to overcome age-related difficulties.

### 4.3. The Effect of Presbycusis on the Audiovisual Integration

As mentioned in the introduction, the studies assessing the effect of presbycusis on audiovisual integration are scarce and have inconsistent results. The studies of Tye-Murray et al. [21] and Reis et Escada [22] assess the effect of presbycusis on visual enhancement produced by speechreading (speech comprehension aided by the visual cues of the face of the speaker). This enhancement of speech comprehension with visual cues is a well-documented phenomenon. It occurs for people with presbycusis, as shown in the studies of Tye-Murray et al. and Reis et Escada, as well as for listeners with normal hearing in a noisy environment [76] or clear and intelligible speech [77,78]. It should be noted that this enhancement can easily be assimilated into the audiovisual integration abilities, but its mechanism seems subtle. Indeed, according to models of speech perception with lipreading, this speech enhancement also relies on the ability to lipread, i.e., the capacity to understand speech with only the visual cues of the speaker’s face (without auditory cues), and to encode the auditory information [21]. The study of Tye-Murray did not find better enhancement given by speechreading for a presbycusis group compared to an age-matched group of normal listeners. They conclude that age-related hearing loss does not lead to better audiovisual integration. On the contrary, the study of Reis and Escada found greater enhancement by speechreading for the group with presbycusis than the age-matched group of normal listeners, which could then arise from better audiovisual integration. The study of Puschmann et al. [23], using a cross-modal distractor paradigm in a categorization task, found that people with presbycusis make more errors when confronted with a multisensory distractor than normal-hearing people of the same age, and then suggest that presbycusis affects the processing of audio-visual information. Finally, a study by Rosemann and Thiel [24] compared the McGurk illusion [25] in a presbycusis population and a normal-hearing population of the same age. They found that the illusion is more pronounced in presbycusis participants and conclude that age-related hearing loss enhances audiovisual integration abilities.

Based on all these studies, it is hard to obtain a clear view of the effects of presbycusis on audiovisual integration. However, the current study using simple eye movement responses to the target, which are the fastest reaction times in humans, enables more direct evidence for increased audiovisual integration with age-related hearing loss. In previous studies, audiovisual integration effects might have been hidden, for instance, by delays in the speech or manual responses. Eye movements are fast and easy to record and provide a more objective data basis.

This result brings new insights concerning the literature on multisensory integration and aging. The reduced sensitivity of sensory systems induced with aging was one of the main hypotheses explaining the improvement of multisensory integration with age. Thus, the results of the current study are coherent with this hypothesis and further show that the capacity of multisensory integration occurs regardless of age.

### 4.4. The Effect of Selective Attention Loss on Audiovisual Integration

The literature has already highlighted that selective attention was one of the cognitive top-down mechanisms impacting multisensory integration [28]. Indeed, selective attention improves the perception of the task-relevant stimulus and suppresses the perception of the task-irrelevant stimulus. Focused attention on a specific sensory modality decreases cortical activity for the other sensory modalities, leading to a decrease in multisensory integration if an attempted sensory modality arrives [79,80,81]. It is also a well-known phenomenon that selective attention is degraded with age [42,43,44,45].

The influence of selective attention loss on multisensory integration abilities for the elderly has benefits and disadvantages. On one hand, the elderly are more likely to be distracted by incongruent sensory modalities [47,48,82]. On the other hand, they are more receptive to the multisensory integration of congruent stimuli. The loss of selective attention with age is otherwise one of the hypotheses explaining the improvement of multisensory integration with age [28]. The result of the current study confirms the impact of selective attention on multisensory integration and is consistent with this last hypothesis. In the current study, the participants were not informed of the audiovisual condition, and the only instruction they received was to look at the LEDs as rapidly and as accurately as possible. They were therefore more focused on the visual component, and some were not even aware of the presence of the auditory component. Thus, it is possible that the participants with greater selective attention capacity were more focused and did not integrate the auditory component.

## 5. Conclusions

The present study introduces a novel research field on the influence of physiologic age-related hearing loss and the audiovisual integration capacities evaluated with saccades and vergence eye movement to visual vs. audiovisual targets using a pure tone. The eye movement results alone show that aging effects are specific to some parameters of eye movements, namely the latency, while velocity or amplitude do not change drastically (see Table 1). The major results are (i) an improvement of audiovisual integration for convergences with age within the elderly group and (ii) an improvement of audiovisual integration for saccades with the deterioration of hearing and selective attention, independently of age.

As convergence is the more complex eye movement, the fact that audiovisual integration improves within the elderly group highlights multisensory compensatory mechanisms that can be mobilized, particularly for perception and action towards targets dependent on depth (near vs. far space).

The improvement of audiovisual integration of saccades with the loss of selective attention as measured by the Stroop test confirms the influence of top-down attentional mechanisms on the quality of multisensory integration.

We suggest that the evaluation of audiovisual integration via eye movement tests could be relevant and helpful for orienting auditory patient rehabilitation. If neuroplasticity and regeneration of hair cells decreasing with presbycusis are not possible, multisensory integration could be of great importance for the quality of audition and quality of life.

## Figures and Tables

**Figure 1 brainsci-12-00591-f001:**
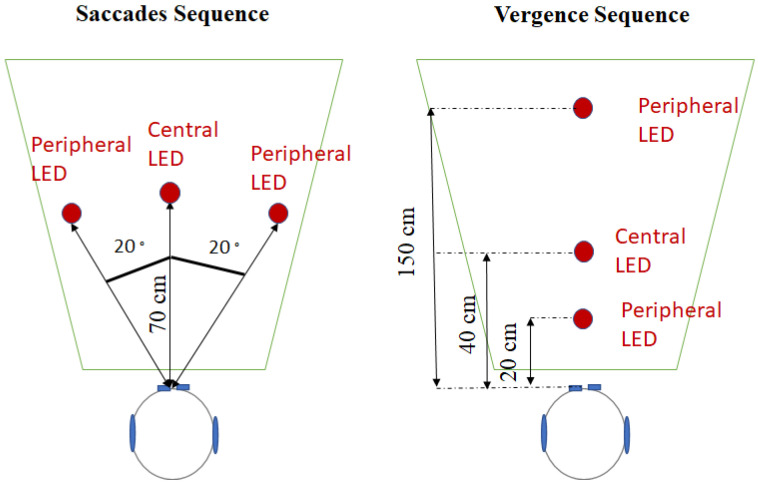
Top-view of the position of the LEDs for the saccades test (**left**) and for the vergence test (**right**).

**Figure 2 brainsci-12-00591-f002:**
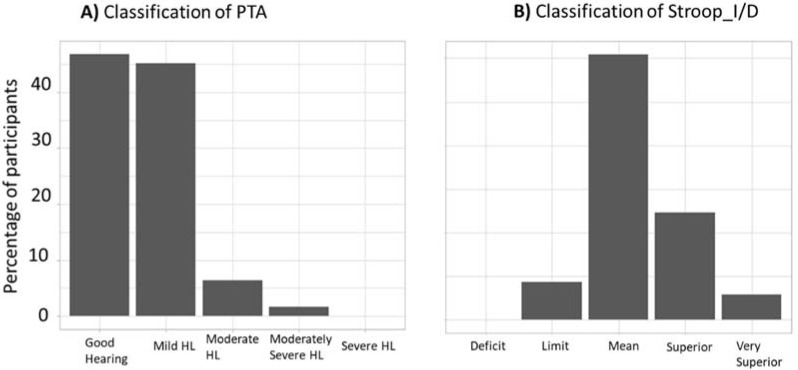
Hearing loss (HL) and Stroop score characterization of EG. (**A**) Classification of the PTA according to the WHO scale, for EG. (**B**) Classification of the Stroop_I/D according to the model built in the study of Bayard et al. [57], for elderly participants. This model allows the categorization of the Stroop_I/D score as a function of the participant’s age above 50 years. The score can be classified into five categories: “deficit”, “limit”, “mean”, “superior”, and “very superior”.

**Figure 3 brainsci-12-00591-f003:**
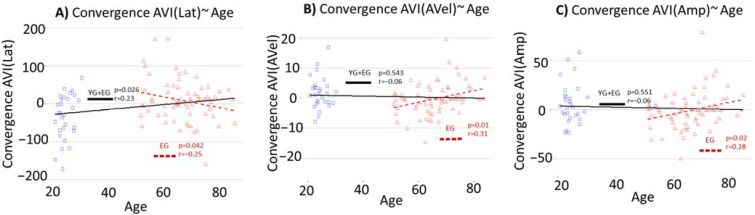
Correlations and regression lines between convergence AVI and age, for (**A**) AVI(Lat), (**B**) AVI(AVel), and (**C**) AVI(Amp), regarding the whole population (young and elderly groups) and elderly group alone. Blue squares represent the young participants; red triangles represent the elderly participants. The black solid line represents the regression line for the whole population and the red dashed line represents the regression line for the elderly group. “r” represents the Pearson correlation coefficient and “*p*” represents the significance of the slopes of the regression line.

**Figure 4 brainsci-12-00591-f004:**
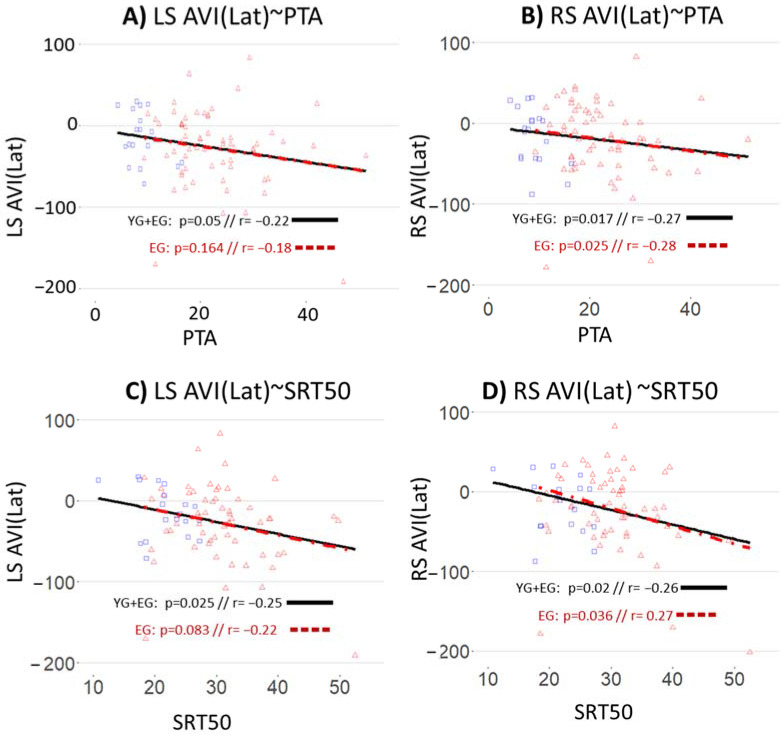
Correlations and regression lines between AVI(Lat) for saccades and hearing tests (PTA and SRT50), for the whole population (YG + EG) and for the elderly participants alone (EG). The left column (**A**,**C**) is for Left Saccade (LS); the right column (**B**,**D**) is for Right Saccade (RS); the first line is for the PTA; the second line is for SRT50. Blue squares represent the participants of the YG; red triangles represent the participants of the EG. The black solid line represents the regression line for the whole population (YG + EG) and the red dashed line represents the regression line for the elderly participants alone (EG). “r” represents the Pearson correlation coefficient and “*p*” represents the significance of the slope of the regression line.

**Figure 5 brainsci-12-00591-f005:**
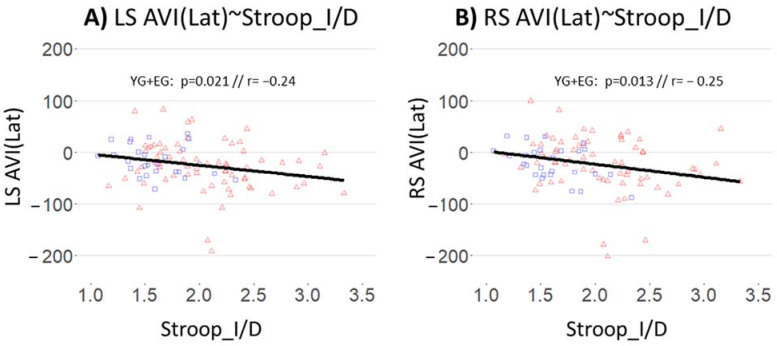
Correlations and regression lines between saccades AVI(Lat) and Stroop_D/I, for (**A**) Left Saccade and (**B**) Right Saccade, regarding the whole population (YG + EG). Blue squares represent the participants of the YG; red triangles represent the participants of the EG. The black solid line represents the regression line for the whole population (YG + EG). “r” represents the Pearson correlation coefficient and “*p*” represents the significance of the slope of the regression line.

**Table 1 brainsci-12-00591-t001:** Eye movement characteristics depending on the group. Means and sd.

		Lat	Pvel	Avel	Amp
D	YG	309 (52)	74 (27)	19 (4)	62 (18)
	EG	371 (62)	78 (25)	15 (4)	47 (15)
C	YG	307 (57)	69 37)	23 (5)	67 (20)
	EG	360 (62)	75 (35)	20 (6)	52 (21)
LS	YG	245 (32)	341 (69)	83 (7)	93 (8)
	EG	298 (55)	318 (85)	78 (8)	88 (8)
RS	YG	255 (35)	346 (72)	84 (6)	94 (6)
	EG	307 (57)	338 (93)	78 (8)	89 (10)

**Table 2 brainsci-12-00591-t002:** Hearing characteristics depending on the group. Means and sd.

	PTA	SRT50	SND50
YG	9.0 (3.0)	21.6 (4.2)	−11.8 (2.9)
EG	22.8 (8.7)	31.2 (8.3)	−10.4 (2.7)

**Table 3 brainsci-12-00591-t003:** Relationships between the AVI scores and age, from simple linear regressions analyses.

		AVI(Lat)~Age			AVI(PVel)~Age			AVI(Avel)~Age			AVI(Amp)~Age		
Movement		Intercept	a	cor	Intercept	a	cor	Intercept	a	cor	Intercept	a	cor
D	For EG + YG	−23.808	0.352	0.136	8.657	−0.19	−0.147	0.694	−0.008	−0.041	6.083	−0.075	−0.1
	For EG	18.003	−0.242	−0.041	−11.922	0.099	0.036	−2.184	0.034	0.08	−9.33	0.145	0.096
C	For EG + YG	−41.214	**0.695 ***	0.226	−2.929	0.142	0.085	1.27	−0.016	−0.062	4.555	−0.058	−0.061
	For EG	126.017	**−1.748 ***	−0.245	14.389	−0.108	0.026	−13.029	**0.193 ***	0.307	−41.297	**0.613 ***	0.279
LS	For EG + YG	−13.377	−0.194	0.090	−4.353	0.218	0.08	−0.47	0.021	0.08	−1.052	0.032	0.106
	For EG	−27.334	0.001	0.000	23.981	−0.199	−0.032	0.236	0.012	0.018	4.069	−0.042	−0.057
RS	For EG + YG	−15.771	−0.097	−0.042	5.309	−0.048	−0.015	1.011	−0.001	−0.004	0.325	0.012	0.037
	For EG	19.101	−0.62	−0.099	−37.628	0.588	0.069	−2.712	0.055	0.074	−0.455	0.025	0.031
“*”: 0.01 < *p* < 0.05													

**Table 4 brainsci-12-00591-t004:** Relations between AVI(Lat) and hearing.

		AVI(Lat)~PTA			AVI(Lat)~SRT50			AVI(Lat)~SND50		
Movement		a	StdError	*t*-value	a	StdError	*t*-value	a	StdError	*t*-value
D	For EG + YG	−0.011	0.837	−0.013	0.655	0.841	0.778	0.377	2.302	0.164
	For EG	−0.298	0.829	−0.359	0.632	0.874	0.724	0.673	2.568	0.262
C	For EG + YG	0.319	0.99	0.322	0.845	0.996	0.848	−2.937	2.808	−1.046
	For EG	1.111	0.998	1.114	1.631	1.045	1.561	−3.431	3.137	−1.094
LS	For EG + YG	**−1.276.**	0.708	−1.802	**−1.478 ***	0.71	−2.081	0.527	1.85	0.285
	For EG	**−1.361.**	0.8	−1.701	**−1.780 ***	0.834	−2.133	0.145	2.272	0.064
RS	For EG + YG	**−2.056 ****	0.755	−2.722	**−1.853 ***	0.77	−2.407	1.224	2.04	0.6
	For EG	**−1.859 ***	0.857	−2.17	**−1.827 ***	0.911	−2.006	0.4	2.55	0.157
“.”: 0.05 < *p* < 0.1; “*”: 0.01 < *p* < 0.05; “**”: 0.001 < *p* < 0.01										

**Table 5 brainsci-12-00591-t005:** Relations between AVI(Lat) and Stroop scores.

		AVI(Lat)~Stroop_I/D			AVI(PVel)~Stroop_I/D			AVI(AVel)~Stroop_I/D			AVI(Amp)~Stroop_I/D		
Movement		a	StdError	*t*-Value	a	StdError	*t*-Value	a	StdError	*t*-Value	a	StdError	*t*-Value
D	For EG + YG	7.101	13.323	0.533	4.006	7.225	0.554	−1.52	1.318	−1.153	−2.12	4.88	−0.434
	For EG	−1.36	13.423	−0.101	4.928	8.756	0.563	−1.552	1.377	−1.127	−2.362	5.017	−0.471
C	For EG + YG	−15.334	15.781	−0.972	6.247	11.935	0.523	0.355	1.787	0.199	1.346	6.552	0.205
	For EG	−14.084	16.207	−0.869	2.653	14.153	0.187	−0.076	1.854	−0.041	−0.601	6.6	−0.091
LS	For EG + YG	**−21.831.**	11.307	−1.931	6.7	15.293	0.438	0.59	1.6	0.369	0.813	1.797	0.452
	For EG	−21.625	12.948	−1.67	6.432	14.719	0.437	0.742	1.754	0.423	0.88	1.93	0.456
RS	For EG + YG	**−25.842 ***	12.271	−2.106	−11.902	18.094	−0.658	1.6	1.8	0.889	1.556	1.845	0.843
	For EG	−21.045	14.139	−1.488	−18.819	21.155	−0.89	1.554	1.915	0.811	1.394	1.97	0.707
“.”: 0.05 < *p* < 0.1; “*”: 0.01 < *p* < 0.05													

## Data Availability

Not applicable.

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
