# Peer review of "Audiovisual Integration for Saccade and Vergence Eye Movements Increases with Presbycusis and Loss of Selective Attention on the Stroop Test"

_brainsci, 2022, doi:10.3390/brainsci12050591_

Round 1

Reviewer 1 Report

  1. Introduction provided sufficient context of background literature; however, there is no theoretical foundation and no clear hypothesis. Therefore, once the reader gets to the results section there is an overwhelming amount of information but it’s not clear how the analysis relates to the research question.
    1. Was there an expectation that saccade (or vergence) amplitude or peak/average velocity would be affected by the presence of an audio input? Also, why were saccades and vergence chosen to probe the integration? Were difference in AVI expected between these movements?
    2. What hypothesis was tested by conducting a regression analysis across the whole sample vs elderly group only?
    3. It’s not clear what the Stroop task adds to this research
  2. The results section should include the performance measures for each group tested, ie, mean latency and other kinematic measures for each group, average scores on the hearing tests, which tests were used to define hearing loss, how many individuals had hearing loss?
  3. The rationale for using a regression analysis for all the different oculomotor measures and age is missing – perhaps this is because there is no hypothesis. I am not convinced that the regression analysis shows any important findings. It’s difficult to appreciate how a slope different from 0 conveys anything meaningful about AVI.
  4. It appears that Figure 3 shows results directly relevant to the research question posed by the study. The paper would improve greatly by refocusing on the main question and conducting only the analyses specific to answering the question (guided by a hypothesis)
  5. If the study had any exclusion criteria, please add to the methods section. Also, it would be informative to provide more information about the visual status of patients. A questionnaire was mentioned, what sort of screening questions were used? What was the acuity range? Did the participants have any visual or ocular conditions? Were any participants excluded based on the questionnaire?
  6. It is necessary to provide more information about the eyetracker, such as sampling frequency, tracking mode (pupil vs pupil/corneal), calibration procedure, calibration/validation error
  7. Please explain why the presentation of the visual and auditory targets was no simultaneous. According to the manuscript, the sound was presented 50 ms before the LED
  8. Lines 207-209 mentioned a difference signal, this requires more explanation. Why were saccades detected based on the difference signal? – Perhaps this is a misunderstanding? Please outline the analysis clearly for both saccades and vergence. Include the details of data processing including any filter applied to the raw data. Were the trials screened by a human observer after the algorithm was applied to removed the outliers? How many trials were excluded based on each criterion?
  9. The results section is cluttered with Tables showing non significant results, consider moving this material to an Appendix of supplementary material

Author Response

Thank you very much for all your helpful comments.

Your whole report is resumed bellow, and we add the answer point by point  (comments in blue and responses in black).

Also, we highlight the change in the new manuscript with green color and comments.

Reviewer 2 Report

Audiovisual Integration for Saccade and Vergence eye Movements Increases With Presbycusis and Loss of Selective Attention on the Stroop Test  By: Martin Chavant  and Zoï Kapoula    

This manuscript deals with the audiovisual integration of information processing in human beings. In particular addressed the study of the effect of presbycusis and its possible influence in multisensory integration with aging. The authors proposed a series of experiments where after checking auditory and visual capabilities they implement a way to check the response associated with the integrated processing. In addition, the authors proposed a check for certain aspects of the attention and inhibition capacities.   

The results obtained by the author seem to indicate that there is an improvement on the audiovisual integration with aging in people with presbycusis and a loss of attention/inhibition capabilities.   

The manuscript is well organized and the way of presenting the information is good. The variables defined to evaluate the audiovisual sensory integration are interesting and, in my view, they allow to grasp a good part of the behavioral response of the subjects. The devices used along the experiments seems to be adequate and with the precision enough for the purpose of the studies proposed. This study could be supplemented using event related potentials, I wonder if the author considered doing that?   

The conclusions of the work presented are interesting and I believe they contribute to a corpus of knowledge that needs to be built. The manuscript presents enough valuable information to the standard of Brain Science and in my view it deserves to be published.   

Some issues of the introduction need to be addressed:

1) In line 10  "The audiovisual integration is probably the kind of multisensory integration the most used, as vision and hearing are the two senses giving the most important amount of information for humans." This should be paraphrased having in consideration that tactil information integrated with vision is surely more significant than audiovisual.   

2) Line 33 "Presbycusis well-known and logical consequences......" is unclear and should be rewritten appropriately.   

3) On line 39 "Thus, consequences of presbycusis appear to a much greater extent than just on hearing issues and seem to affect more central cortical aspects." This is a too strong affirmation, there must be a large variety of reasons  for the possible changes in cortical areas in co-occurrence of presbycusis, but not proving causality.  

4) On line 48 and 49 "The audiovisual integration (AVI) is probably the kind of multisensory integration the most performed, as hearing and vision are for humans the most important senses to perceive the current world." As mentioned in relation to line 10. Tactile information is highly relevant and MI between tactile and vision is probably as important or more than audiovisual. So, in my view this phrase should be modified.   

Author Response

Thank you very much for all your helpful comments.

We have taken into account all your comments. Changes in the manuscript are indicated in green, with comments.
